# Effect of Gibberellic Acid on Production of Biomass, Polyphenolics and Steviol Glycosides in Adventitious Root Cultures of *Stevia rebaudiana* (Bert.)

**DOI:** 10.3390/plants9040420

**Published:** 2020-03-30

**Authors:** Ashfaq Ahmad, Haider Ali, Habiba Khan, Almas Begam, Sheraz Khan, Syed Shujait Ali, Naveed Ahmad, Hina Fazal, Mohammad Ali, Christophe Hano, Nisar Ahmad, Bilal Haider Abbasi

**Affiliations:** 1Centre for Biotechnology and Microbiology, University of Swat, Swat 19200, Pakistan; sayyadashfaq111@gmail.com (A.A.); akhaider48@gmail.com (H.A.); habiba.khanbiotech@gmail.com (H.K.); almaskhadim1993@gmail.com (A.B.); sherazkhan@aup.edu.pk (S.K.); shujaitswati@uswat.edu.pk (S.S.A.); alimoh@uswat.edu.pk (M.A.); 2Department of Horticulture, The University of Agriculture, Peshawar 25120, Pakistan; naveedhorticons@gmail.com; 3Pakistan Council of Scientific and Industrial Research (PCSIR) Laboratories Complex, Peshawar 25120, Pakistan; hina_fazalso@yahoo.com; 4Université d’Orléans, Laboratoire de Biologie des Ligneux et des Grandes Cultures (LBLGC), INRA USC1328, F28000 Chartres, France; christophe.hano@univ-orleans.fr; 5Department of Biotechnology, Quaid-i-Azam University Islamabad, Islamabad 45320, Pakistan

**Keywords:** *Stevia rebaudiana*, gibberellic acid, steviol glycosides, polyphenolics, antioxidant activity

## Abstract

In current study, the effect of gibberellic acid was tested for production of biomass, polyphenolics and Steviol glycosides in adventitious root cultures of *Stevia rebaudiana*. Adventitious cultures were induced from the roots of in vitro grown plantlets on Murashige and Skoog (MS) medium containing combination of gibberellic acid (GA_3_; 0.5, 1.0, 1.5 and 2.0 mg/L) and naphthalene acetic acid (NAA; 0.5 mg/L). Initially, a known mass of inoculum roots were shifted into suspension media augmented with various GA_3_ concentrations. The growth behavior of adventitious roots was recorded every 3 days for a period of 30 days. Maximum biomass biosynthesis (13.12 g/flask) was noticed in exponential phase on 27th day in the suspension containing 2.0 mg/L of GA_3_. Other GA_3_ concentrations also displayed optimum patterns of biomass accumulation as compared to the control. Adventitious roots were investigated for total phenolic content (TPC) and production (TPP), total flavonoid content (TFC) and production (TFP), and 1, 1-diphenyl-2-picrylhydrazyl (DPPH)-based antioxidant potential. Maximum phenolics (TPC 9.84 mg gallic acid equivalent (GAE)/g-dry weight (DW)) and TPP (147.6 mg/L), TFC (5.12 mg Quercitin equivalent (QE)/g-DW) and TFP (76.91 mg/L) were observed in 2.0 mg/L GA_3_ treated cultures. The same concentration of gibberellic acid enhanced antioxidant activity (77.2%). Furthermore, maximum stevioside (7.13 mg/g-DW), rebaudioside-A (0.27 mg/g-DW) and dulcoside-A (0.001 mg/g-DW) were observed in roots exposed to 2.0 mg/L GA_3_. This is the first report on the application of GA_3_ on biomass accumulation and secondary metabolite production in *S. rebaudiana*. The current study will be helpful to scale up the adventitious root cultures in bioreactors for the production of biomass and pharmaceutically important secondary metabolites.

## 1. Introduction

*Stevia rebaudiana* is one of the important therapeutic and sweetest plants worldwide, which belongs to family Asteraceae [1]. South America’s indigenous tribes used it since ancient times and called it “kaa-hee” (“sweet herb”). It is a wild plant and has its origin in the highlands of Northeastern Paraguay [2,3]. Among other members of the family Asteraceae, only *S. rebaudiana* displays the highest level of sweetness [4]. Currently, it is cultivated throughout the world for its stevioside (Stev.) and rebaudioside-A (Reb. A) content [5,6,7]. *S. rebaudiana* is cultivated in many countries, including Tanzania, Indonesia, Korea, Thailand, Singapore, Malaysia, Australia, Argentina, Russia, India, Abkhazia, China, Mexico, Brazil, Paraguay, Taiwan, Canada, Japan and the United States of America (California and Hawaii) [8,9,10,11]. It has been used for treating heartburn, obesity, hyperglycemia and dental caries [12]. Moreover, it also enhances glucose tolerance [13]. Being an anti-hyperglycemic agent, it stimulates the pancreas to secrete insulin [14,15]. Due to no receptors for the absorbance of Stevioside, it is used for the treatment of diabetic patients because these contents cannot enter into the blood stream.

Conventional propagation of *S. rebaudiana* occurs by cuttings and seeds [16]. There are various factors that affect the traditional cultivation, such as nutrient uptake, land availability, pests and weather, which may negatively affect the major and minor compounds produced in *S. rebaudiana* [17]. Plant cell and tissue culture is an alternative to these methods, which provides a very useful technique through which a large number of clones of *S. rebaudiana* can be obtained. In order to produce biomass and secondary metabolites, the culturing of adventitious roots is considered to be more reliable than other culturing techniques. The reason is that adventitious root culture can be easily scaled up; additionally, adventitious root cultures make it easy to control the physical and chemical conditions [18]. The accumulation and production of bioactive compounds mostly take place in roots of various plants. 

Plants are a source of many important bioactive compounds. Currently, about 100,000 compounds have been identified in plants and about 4000 new compounds are discovered each year [19]. A wide variety of chemicals have been obtained from plants, including ginsenosides, paclitaxel, echinacosides, shikonin, berberine and hypericin [20,21,22,23]. These compounds are involved in the elimination of toxic free radicals, enzyme activation, activation of immune system, controlling the expression of genes and cell death initiation [24,25]. These bioactive compounds also function as nutraceuticals, to cure cardiac diseases, flavor foods, provide antioxidants, create potential drugs, supply pharmaceuticals and act as anticarcinogens. 

It is a common phenomenon that the addition of multiple physical and chemical elicitors to the culture media modulates the biosynthesis of biomass and secondary metabolites (SM) [26]. One of the ways to enhance the production of SM is elicitation [27]. Elicitors are agents that stimulate secondary metabolism in plants, as it is believed that when a pathogenic organism attacks plants, the plants will react by activating the defense system and ultimately release secondary metabolites. Similarly, the plant cells produce the same response when challenged by an elicitor and activate the defense system [28]. Nonetheless, the exact mechanism of elicitation is poorly understood. In the current study, gibberellic acid (GA_3_) was used as elicitor; it is a hormone, acting as regulator of stem and cells elongation, and fruit development [29]. Gibberellic acid is synthesized in the same pathway used for artemisinin biosynthesis, and is a diterpenoid [30,31,32]. Various studies have been conducted on the use of GA_3_ as elicitor. GA_3_ was used to enhance the production of caffeic acid derivatives and tanshinones in hairy root cultures of *Echinacea pupurea* and *Salvia miltiorrhiza* [33,34]. In another study, GA_3_ was used in *Artemisia annua* and it boosted the amount of artemisinin [35,36,37], which used GA_3_ as an elicitor that enhanced the biosynthesis of polyphenolics (flavonoids and phenolics) in suspended cells of Artemisia.

## 2. Results

### 2.1. Effect of Gibberellic Acid on Adventitious Roots Biomass and Growth Kinetics

Adventitious root culture of *S. rebaudiana* was established. The data regarding biomass accumulation and growth kinetics of adventitious roots were recorded for 30 days with an interval of 3 days. Initially, 0.233, 0.231, 0.189, 0.410 and 0.139 g of adventitious roots inoculum were inoculated into T1, T2, T3, T4 and a control (Table 1) liquid mediums, respectively, containing constant naphthalene acetic acid (0.5 mg/L NAA; *Phyto* Technologies Laboratories, USA) and different concentrations of GA_3_ (0.5, 1.0, 1.5, 2.0 mg/L). A shorter lag phase was observed in all treatments, ranging from day 3 to day 9, with very little increase in biomass. This was followed by a prolonged log phase that ranged from day 12 to day 27; during this phase, a higher increment in biomass accumulation was observed than lag phases. The log phases were followed by stationary phases, which ranged from day 27 to day 30. The augmentation of 2.0 mg/L of gibberellic acid (GA_3_; *Phyto* Technologies Laboratories, USA) to the culture media promoted maximum biosynthesis of fresh biomass (13.12 g/flask) on 27th day of culture incubation (Figure 1). However, all the GA_3_ treated cultures showed better results than the control culture (3.2 g/flask) which lacked GA_3_. From these results, it was clear that the biomass accumulation in liquid media showed dependency on plant growth regulators (PGRs). It was observed from the growth curve that the biosynthesis of fresh biomass displayed maximum accumulation on day 27 of growth kinetics, but a decline pattern in biomass biosynthesis was noted between 28–30 days (Figure 1). In this study, 0.5 mg/L augmented cultures displayed a 15-fold increase in biosynthesis of fresh biomass; however, the higher concentrations (1.0 and 1.5 mg/L) exhibited 32- and 38-fold increment in biosynthesis of fresh biomass on 27th day of growth kinetics. The highest concentration of GA_3_ (2 mg/L) produced a similar amount of biomass (32-fold), similar to 1 mg/L GA_3_. Therefore, it was concluded that 1.5 and 2.0 mg/L GA_3_ ensures the best elicitors for adventitious root production in *S. rebaudiana*.

The entire GA_3_ treated cultures showed better results than the control culture (3.2 g/flask) that lacked GA_3_. From these results it was clear that the biomass accumulation in liquid culture depends on GA_3_. 

### 2.2. Effect of Gibberellic Acid on Polyphenolics and Antioxidant Activity

In this experiment, multiple 0.5, 1.0, 1.5, 2.0 mg/LL GA_3_ concentrations were investigated for the biosynthesis of phenolics content. We found that GA_3_ (2.0 mg/L) exhibited 9.84 mg gallic acid equivalent (GAE)/g-DW (dry weight) phenolics biosynthesis, 1.5 mg/L of GA_3_ displayed 7.80 mg GAE/g-DW, 1.0 mg/L of GA_3_ resulted 6.36 mg GAE/g-DW and 0.5 mg/L of GA_3_ displayed 7.48 mg GAE/g-DW of total phenolics content (TPC) on the 30th day of growth kinetics (Figure 2). In this study, the highest values (9.84 mg GAE/g-DW) were obtained with 2.0 mg/L of GA_3_ as compared to the control (7.22 mg GAE/g-DW). Furthermore, 1.0 mg/L of GA_3_ was found to be inhibitory among the other tested concentrations, while 2.0 mg/L was found to be the most stimulating concentration. A similar pattern of phenolic production was also observed for different concentrations of GA_3_. In this study, the application of 0.5 mg/L of GA_3_ displayed poor phenolics production while 2.0 mg/L of GA_3_ positively encourage phenolics production, as shown in Figure 2.

Currently, total phenolic content (TPC) and total phenolic (TP) production was dependent on dry biomass as well as on PGR in *S. rebaudiana* adventitious root cultures. Herein, the higher concentration of GA_3_ (2.0 mg/L) was found effective for biosynthesis of dry biomass (0.20-g/L), as well as for biosynthesis of phenolics (TPC; 9.84 mg GAE/g-DW) in comparison to control treatment (Figure 2). It means that dry biomass biosynthesis; total phenolic production (TPP) and phenolics (TPC) in a suspension culture of *S. rebaudiana* were found dependent on plant growth regulators (Figure 2). This ultimately proves that TPC and TPP are dependent on both biomass and plant growth regulators. The TPC in other cultures was also in correlation with dried root biomass (DRB) as well as PGRs. These results clearly indicated the addition of a higher concentration of GA_3_ (2.0 mg/L) is the most suitable candidate for maximum biosynthesis of phenolics content and phenolics production in *S. rebaudiana* adventitious root cultures in vitro. 

In this study, total flavonoid content (TFC) and total flavonoid production (TFP) increased with the increasing concentrations of GA_3_. Cultures treated with 0.5, 1.0 and 1.5 mg/L of GA_3_ resulted in 4.20, 4.26 and 4.90 mg QE/g-DW flavonoids biosynthesis, but GA_3_ (2.0 mg/L) proved to be effective in flavonoid biosynthesis (5.12 mg QE/g-DW; Figure 3). The highest TFC accumulation was shown by 2.0 mg/L of GA_3_, as compared to the control culture (4.74 mg QE/g-DW on day 30). A lower concentration of GA_3_ produced lesser flavonoid content as compared to higher concentration of GA_3_, i.e., 1.5 and 2.0 mg/L. A similar increasing pattern of flavonoid production was also displayed by various concentrations of gibberellic acid (Figure 3). 

The production of TFC and TFP were also found to be DRB as well as PGRs dependent. There was a good correlation of TFC and TFP with DRB and PGR. Maximum TFC (5.12 mg QE/g-DW) and maximum DRB (0.20 g/L) were displayed by the roots exposed to a higher concentration of GA_3_ (2.0 mg/L), which was comparatively higher than the control treatment. This shows that as PGRs concentration and biomass increases, the TFC also increases. This clearly indicates that TFC correlates with dry biomass as well as PGR. The production of flavonoid was also found to be dry biomass and PGRs dependent (Figure 3).

In current experiment, GA_3_ treated cultures were investigated through radical scavenging activity (RSA) for DPPH-based antioxidant potential. Here, 0.5 mg/L showed 73.6% of the activity, 1.0 mg/L showed 74.5% activity, 1.5 mg/L showed 76.8% activity, while 2.0 mg/L of GA_3_ showed the highest activity of 77.2% (Figure 4). This indicated that the activity increased with increasing concentration of GA_3_. The lowest activity was shown by 0.5 mg/L of GA_3_, (Control: 73.8% of RSA activity). Radical scavenging activity (RSA) displayed a linear correlation with flavonoid and phenolic content and polyphenolic production in *S. rebaudiana* root cultures in vitro (Figure 4 and Figure 5). Lowest RSA and TFC, TFP, TPC, TPP were shown by 0.5 mg/L of GA_3_, while the highest RSA and TFC, TFP, TPC and TPP were shown by 2.0 mg/L of GA_3_. This showed that the RSA is phenolic and flavonoid dependent, as the content of phenolics and flavonoids increase, the RSA also increases, and vice versa. 

### 2.3. Effect of Gibberellic Acid on Steviol Glycosides

In this study, steviol glycosides (steviosides, rebaudioside-A and dulcoside-A) were investigated in adventitious root cultures of Stevia exposed to various concentrations of GA_3_ (Figure 6). Herein, the maximum stevioside content (7.13 mg/g-DW) was observed in roots exposed to 2.0 mg of GA_3_, while 3.39 mg/g-DW was observed in control treatment. Other concentrations of GA_3_ (0.5, 1.0 and 1.5 mg/L) were found less effective to improve stevioside content in adventitious root cultures in Stevia. Furthermore, it was observed that the control treatment displayed maximum rebaudioside-A content, which is >6 mg/g-DW. However, the higher concentration of GA_3_ (2.0 mg/L) exhibited >5 mg/g-DW biosynthesis of rebaudioside-A content. The other treatments (0.5, 1.0 and 1.5 mg/L GA_3_) showed lower production of rebaudioside-A content than the control and 2.0 mg/L GA_3_. In this study, the biosynthesis of dulcoside-A content was comparatively lower than stevioside and rebaudioside-A content in adventitious root cultures of Stevia exposed to multiple concentration of GA_3_. Here, 2.0 mg/L GA_3_ produced >1 mg/g-DW of dulcoside-A content, which is greater than the control and other GA_3_ treatments. These results suggest that 2.0 mg/L GA_3_ was found effective for the overall biosynthesis of Steviol glycosides.

## 3. Discussion

These results are in agreement with various other studies. The data regarding the effect of GA_3_ on the biosynthesis of metabolites of interest and biomass in *S. rebaudiana* adventitious roots cultures are not available in literature, but its effect on some other plants has been studied. In hairy roots of *E. purpurea*, the application of GA_3_ positively enhanced dry biomass accumulation [33]. The stimulating effect of GA_3_ on biomass accumulation was also found in another study [38] that found that GA_3_ enhanced the growth of hairy roots of Datura when used in the concentrations ranging from 0.1 to 0.0001 mg/L. However, not all plant species respond the same way to GA_3_, nor different clones of the same plant [38,39,40], which study also found a substantial effect of GA_3_ on cell size as well as root growth in maize. Ali et al. [37] conducted a study on using GA_3_ as elicitor; they found that GA_3_ resulted in a 2.4-, 2.6- and 2.5-fold increment in biosynthesis of fresh biomass when cultures were exposed to 2.0, 1.0 and 0.5 mg/L, respectively. Smith et al. [35] used six different concentrations of GA_3_ for determining its effect on the accumulation of biomass in hairy root cultures of *A. annua.* The data revealed that GA_3_ at 0.01 mg/L (28.9 µM) enhanced the hairy roots’ growth rate by 24.9%. GA_3_ affect the hairy root cultures of numerous plant species [33,38]. Moderate concentrations of GA_3_ were found to enhance the biomass accumulation in hairy root cultures of economical plants [35], as well as organogenesis in potted *Zea mays*. Similar results were found in another study in which GA_3_ highly enhanced the artemisinin content as well as biomass in hairy root cultures of Artemisia [35].

These results are in agreement with other studies in which GA_3_ was used to enhance the biosynthesis of caffeic acid derivatives and tanshinones in *S. miltiorrhiza* and *E. purpurea* hairy root cultures, respectively [33,34]. In another study, GA_3_ enhanced the accumulation of artemisinin in *A. annua* [35,36]. Furthermore, GA_3_ was also found to boost the proliferation of hairy roots and biosynthesis of phenolics in *S. miltiorrhiza.* Ali et al. [37] conducted a study by using GA_3_ as elicitor; they used three concentrations, 2.0, 1.0 and 0.5 mg/L and found that when GA_3_ is used at lower concentration such as 0.5 mg/L, it is inhibitory for TPC, while higher concentrations, i.e., 1.0 and 2.0 mg/L, stimulate the production of TPC. On the other hand, a lower concentration of GA_3_ stimulates the production of TFC and higher concentration inhibited it. Similarly, highest RSA was recorded for 1.0 mg/L, which was followed by 0.5 and 2.0 mg/L of GA_3_. Teszlák et al. [41] used GA_3_ and found that it enhances the anthocyanin content in grape. Banyai et al. [36] studied the effect of exogenous GA_3_ and found that the artemisinin production in GA_3_ treated plants was much higher than in control plants. Alonso-Ramírez et al. [42] studied the effect of GA_3_ in salicylic acid (SA) biosynthesis and found that it enhances its production in Arabidopsis. Abbasi et al. [33] studied *E. purpurae*; they used 0.005 to 1.0 μM (eight different concentrations) of GA_3_ to find its effect on hairy root cultures. They found that the exposure of cultures to 0.025 μM GA_3_ showed the highest accumulation of culture biomass, secondary metabolites and free radical scavenging activity. GA_3_ can alter the production of metabolites in plants such as *Catharanthus roseus* treated with GA_3_; the production and accumulation of antioxidants and ajmalicine was enhanced [43].

In contrast, Khalil et al. [17] observed 10.20 mg/g-DW of stevioside content in Stevia in vitro shoots. The differences in data may be due to the application of polyamines or tissues differences. However, other researchers extracted a lower amount of stevioside content (0.082, 2.9, 0.28, 6.23 mg/g-DW) from in vitro shoots of *S. rebaudiana* [44,45,46,47]. Furthermore, Ladygin et al. [46] detected 0.06 mg/g-DW stevioside content in calli cells. Similarly, Mathur and Shekhawat [5] and Reis et al. [6] observed minimum quantities of stevioside content in cell suspension and adventitous root cultures of *S. rebaudiana*. Moreover, Bondarev et al. [47] detected 4.6 mg/g-DW stevioside content in Stevia in vitro plantlets. Among Steviol glycosides, rebaudioside is sweeter than stevioside, but the production of rebaudioside and dulcoside are limited. Therefore, in the current study, lower contents of rebaudioside-A and dulcoside-A were also observed in response to various GA_3_ treatments.

## 4. Materials and Methods 

### 4.1. Seed Collection and Germination

The nursery grown plants of *S. rebaudiana* were used for the seed collection. These plants were cultivated in the Ground and Garden Nursery, The University of Agriculture Peshawar, Pakistan. Protocol reported by [24] was followed for surface decontamination of seeds. Here, 0.1–0.2% HgCl_2_ and 70% ethanol was exploited as the surface decontaminator for 1 min and then the seeds were repeatedly washed with autoclaved water to remove the toxic content. The dried surface decontaminated seeds were transferred into Murashige and Skoog medium (MS) (*Phyto* Technologies Laboratories, USA) having no PGRs, along with a 30 g/L carbon source (sucrose), solidified with 7–8 g/L agar (Sigma) and the pH level was adjusted to 5.3, followed by media sterilization at 121 degrees Celsius for 20–25 min. A reliable growth chamber with controlled environment (40 mol m^2^/s, 25 ± 2 °C, 16/8 hr) was used for culture incubation. Herein, vigorous roots as explants for the development of adventitious root cultures were obtained after 30 days of incubation in suspension media.

### 4.2. Development of Adventitious Roots Cultures

Root explants of in vitro grown plantlets were shifted to MS–basal media augmented with 0.5 to 2.0 mg/L NAA concentrations in Erlenmeyer flasks (100 mL) containing 33–40 mL culture media. To obtain adventitious root cultures of *S. rebaudiana*, the cultures were placed on rotary orbital shaker with 120 rpm (Gallenkamp, England) at optimum temperature (25 °C) in the dark for 49 days. In preliminary experiments, adventitious roots as inoculum were shifted to MS media augmented with different concentrations of NAA, i.e., 0.5–2.0 mg/L. Adventitious roots emerged from root segments were excised after 3 weeks. Five-tenths mg/L of NAA was found optimum for adventitious root development in liquid cultures as compared to other concentrations. Therefore, constant NAA concentration (0.5 mg/L) was combined with various GA_3_ concentrations, including 2.0, 1.5, 1.0, 0.5 mg/L in liquid media, which was followed by inoculation of adventitious roots into T1, T2, T3, T4 and the control liquid media (Table 1). These treatments were incubated on an orbital shaker at a rate of 120 rpm for a period of 4 weeks; data concerning the kinetics of growth were recorded for a period of 30 days, with a 3 day interval. An accurate growth curve was designed from the growing biomass of roots in liquid media exposed to multiple concentrations of GA_3_.

### 4.3. Investigation of Biomass of Stevia Adventitious Root Cultures

For determining the fresh weight (FW), suspended roots of Stevia were collected from the liquid media. Sterile distilled water was used to wash these roots and then a filter paper was used to remove excess water by pressing gently the roots over it. Finally, the roots were weighed. Likewise, to determine the dry weight (DW) of the roots, the roots were dried in oven at 50 °C for 24 hr and its final weight was measured in grams. The adventitious roots’ FW and DW were indicated as grams per flask.

### 4.4. Biochemical Analysis

#### 4.4.1. Extract Preparation

The dried *S. rebaudiana* adventitious roots were ground using an electrical grinder for obtaining fine powder for the preparation of extract for various biochemical analyses. The extract preparation was carried out according to the method of Fazal et al. [18]. In a sterilized falcon tube, an exact amount of plant materials (10 mg) obtained from multiple treatments was combined with HPLC grade ethanol (10 mL). The combined mixture was vortexed for 2 hours and then centrifuged (15 min at 14,000 rpm). The supernatant layer was taken and was used to determine different activities.

#### 4.4.2. Determination of Total Phenolic and Total Flavonoid Contents

The determination of TPC of each sample was carried out according to the methodology of Fazal et al. [18]. Concisely, Folin–Ciocalteus reagent at concentration of 0.1 mL (2N) was combined with sterile distilled water (2.55 mL) and extract (0.03 mL). The mixture was centrifuged at 10,000 rpm for 14 min before incubation for a period of 10 min, which was followed by filtration in a cuvette of a UV–visible spectrophotometer through 45 μm membranes. The absorbance of the resulting mixture was measured at 760 nm. Various gallic acid concentrations (Sigma; 1.0–10 mg/mL; R2 = 0.9878) were used for plotting the standard calibration curve. The results were obtained as gallic acid equivalent (GAE) mg/g of dry root biomass (DRB) from percent TPC, the equation is as follows: % total phenolic content = 100 × (As − Ab)**/**(CF − DF), where As is the optical density (OD) of treated culture, Ab is the OD of the blank, DF is the dilution factor and CF is the conversion factor from the standard curve. Similarly, for determination of total flavonoid content, the methodology of Fazal et al. [18] was followed. Briefly, 0.25 mL of extract of treated cultures was combined with pure water (1.25 mL; autoclaved) and reaction component (0.075 mL 5% (aluminum chloride, w/v)). This was followed by mixing the solution with 1M of NaOH (0.5 mL), and then the solution was incubated for 10 min, after incubation the solution was centrifuged at 10,000 rpm for 14 min. Likewise, the OD of the combined solution was found at 510 nm using the UV–visible spectrophotometer. 

#### 4.4.3. DPPH-Based Antioxidant Activity

The methodologies of Ahmad et al. [24] and Zamir et al. [48] were followed for determination of DPPH radical scavenging activity (DRSA). The DRSA activity was determined for each treatment of GA_3_. The DRSA needs two stock solution preparations. The first stock solution was the extract preparation from various adventitious roots culture exposed to various GA_3_ concentrations (0.5, 1.0, 1.5, 2.0 mg/L). The dry, powdered adventitious roots (5 mg) exposed to 0.5 mg/L GA_3_ was separately dissolved in 20 mL of HPLC-grade ethanol (Sigma Aldrich, Germany). The solution was centrifuged at 10,000 rpm for 10 min (PV-1; Grant Instruments, UK). The supernatant was collected for further analysis. The same procedure was followed for other treatments of GA_3_. The second stock solution was the DPPH free radicals, in which powdered DPPH free radicals (Sigma Aldrich; 0.25 mg) were dissolved in 20 mL of HPLC-grade ethanol (Sigma Aldrich, Germany). It was necessary to check the OD of the DPPH solution at 517 nm using the spectrophotometer (UV–visible double beam spectrophotometer; London). If the OD of the DPPH solution was greater than 1.0, then it needed further dilution by adding 20 mL of ethanol and again checking its OD. If the OD was below 1.0, then it was ready for antioxidant activity. The final step was the investigation of antioxidant activity. Here, 1 mL solution was taken from stock 1 (GA_3_ treated culture) and combined with 2 mL of stock 2 (DPPH solution) in a spectrophotometer cuvette. To avoid oxidation and to complete the reaction, the combined solution was incubated for 30 min in the dark. After incubation, the absorbance was checked at 517 nm and radical scavenging activity was calculated as percentage of DPPH discoloration using the following equation at 517 nm:

DRSA (%) = 100 × (1 − AP/AD)

where AP represents absorbance of extract at 517 nm and AD is the absorbance of the DPPH solution without tissue extract.

#### 4.4.4. Quantification of Steviol Glycosides in Adventitious Root Cultures

*S. rebaudiana* adventitious root cultures were investigated for stevioside, rebaudioside-A and dulcoside-A content by using the protocol of Aman et al. [44]. To prepare extract for HPLC analysis, 200 mg dried powder from each treatment was dissolved independently in 10 mL HPLC-grade ethanol. The extraction was repeated for three times. After 72 h, the ethanol was evaporated under controlled pressure at 40 °C. After ethanol evaporation, a crude extract was obtained. Exactly 1 mg of extract from each treatment was independently dissolved in 2 mL of ethanol. The solution was filtered through a 0.45 Millipore microfilter. The filtrate was then used for quantification of Steviol glycoside through HPLC. For Steviol glycoside quantification we used a Perkin–Elmer HPLC system (USA). The HPLC conditions included: HPLC equipped with a binary HPLC pump (1525), the column was C18 (ODS) with dimensions 25 × 4.6 mm, 5 μm particle size and 5 mm internal diameter, and a 2487duel λ absorbance detector attached with the integrated Empower 2 software. The analysis was performed at ambient temperature with an injection loop with 10 μL capacities. For the mobile phase, HPLC-grade water (Sigma; A; 25%) and acetonitrile (Sigma; B; 75%) were used. During quantification, a volume of 10 µL was used to inject at a 1.0 mL/min flow rate. Steviol glycoside standard containing stevioside, rebaudioside-A and dulcoside-A, purchased from Sigma (USA) Laboratories, was run at first for standardization of retention time of each Steviol glycoside. Stevioside, rebaudioside-A and dulcoside-A contents were identified in each sample of adventitious root cultures by comparing retention times of samples with the standard. The diterpene glycosides (dulcoside-A, rebaudioside-A and stevioside) were indicated in the results as mg per g on the basis of dry weight.

### 4.5. Formatting of Mathematical Components

To investigate the replicated mean values, we used Statistix software version 8.1 (USA) and then the least significant difference (LSD) was also obtained using the same software. Moreover, the mean values along with standard errors were presented in graphical form through Origin Lab (8.5 version) software.

## 5. Conclusions

This study focused on the elicitation (gibberellic acid) of *S. rebaudiana* adventitious root culture in suspension cultures for biosynthesis of metabolites of interest and biomass accumulation. The adventitious root cultures of *S. rebaudiana* in liquid cultures were exposed to various concentrations of GA_3_ with constant NAA augmentation. In this study, 2.0 mg/L GA_3_ was observed to be the most suitable candidate for biosynthesis of biomass on the 27th day of growth kinetics. Likewise, the same concentration was found effective for the biosynthesis of TFC, TFP, TPC and TP production, as well as DPPH activity. The same GA_3_ concentration improved the production of stevioside, rebaudioside-A and dulcoside-A content. Therefore, there is a need for scaling up these cultures to produce uniform and optimum quantities of metabolites of interest for commercial application in specific bioreactor systems. 

## Figures and Tables

**Figure 1 plants-09-00420-f001:**
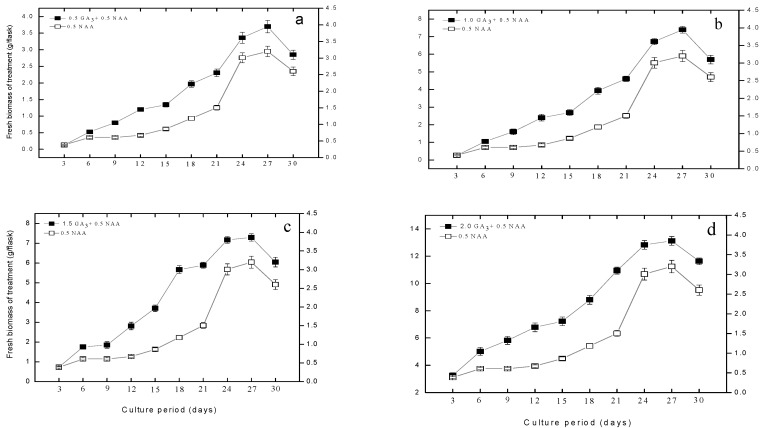
Fresh adventitious roots biomass accumulation in T1 (GA_3_ 0.5 mg/L + NAA 0.5 mg/L), T2 (GA_3_ 1.0 mg/L + NAA 0.5 mg/L), T3 (GA_3_ 1.5 mg/L), T4 (GA_3_ 2.0 mg/L) and control (0.5 mg/L NAA) media. In this experiment, the growth kinetics of *S. rebaudiana* adventitious root cultures was studied after 30 days of culture duration. The triplicate data in the mean along with standard errors (SE) are significantly different at *p* < 0.05.

**Figure 2 plants-09-00420-f002:**
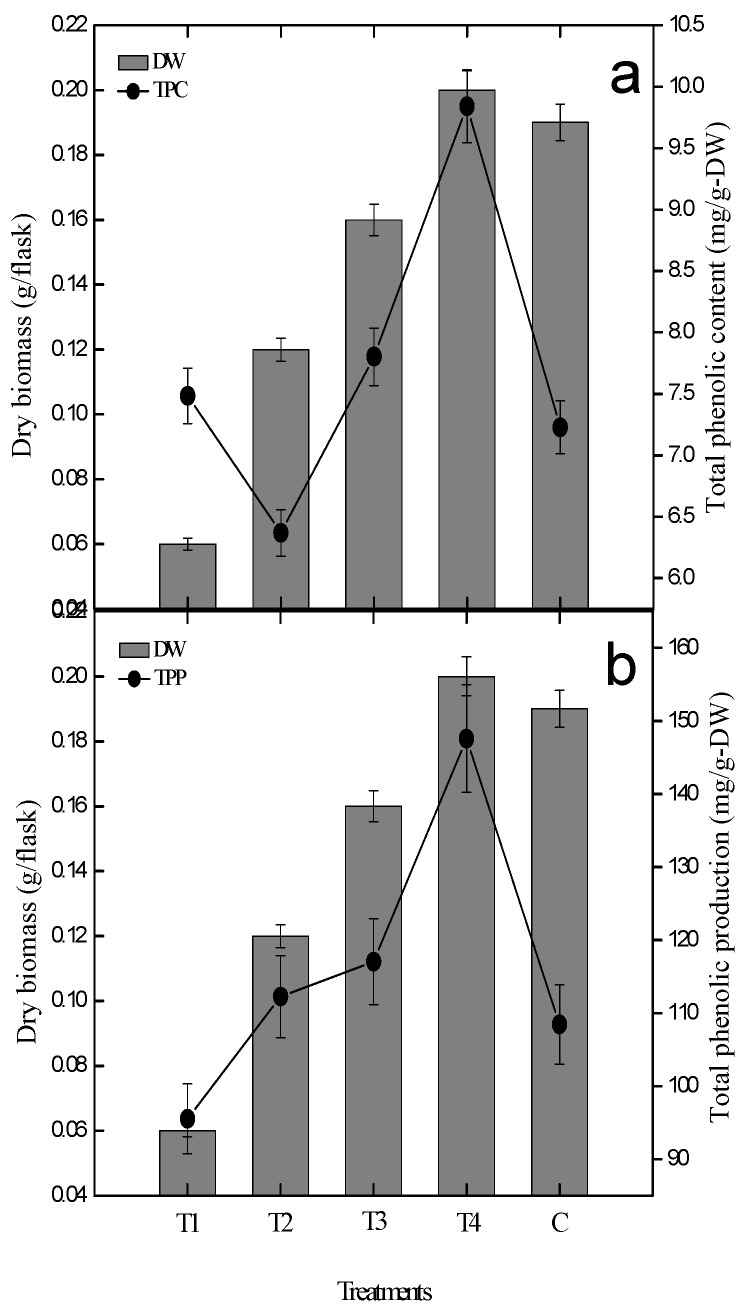
Biomass and PGRs (plant growth regulators) show dependent linear and positive correlation with phenolics biosynthesis (**a**) and phenolics production (**b**) in *S. rebaudiana* adventitious root cultures in vitro. The triplicate data in the mean along with standard errors (SE) are significantly different at *p* < 0.05.

**Figure 3 plants-09-00420-f003:**
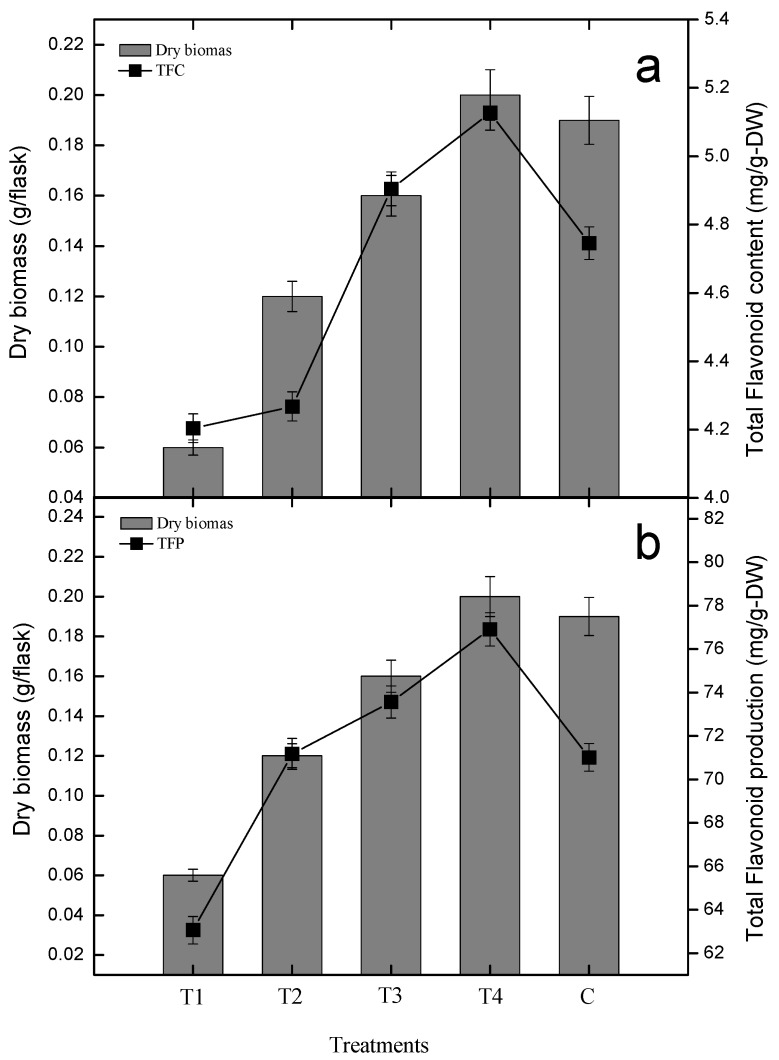
Biomass and PGRs show dependent linear and positive correlation with flavonoids biosynthesis (**a**) and flavonoids production (**b**) in *S. rebaudiana* adventitious root cultures in vitro. The triplicate data in the mean along with standard errors (SE) are significantly different at *p* < 0.05.

**Figure 4 plants-09-00420-f004:**
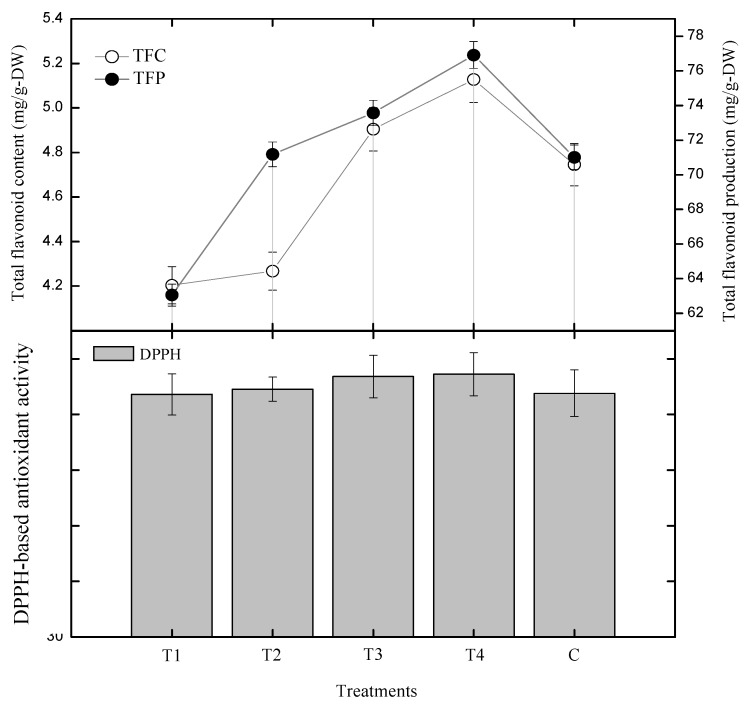
Correlation between DPPH-based antioxidant activity and phenolics in *S. rebaudiana* adventitious root cultures in vitro. The triplicate data in the mean along with standard errors (SE) are significantly different at *p* < 0.05.

**Figure 5 plants-09-00420-f005:**
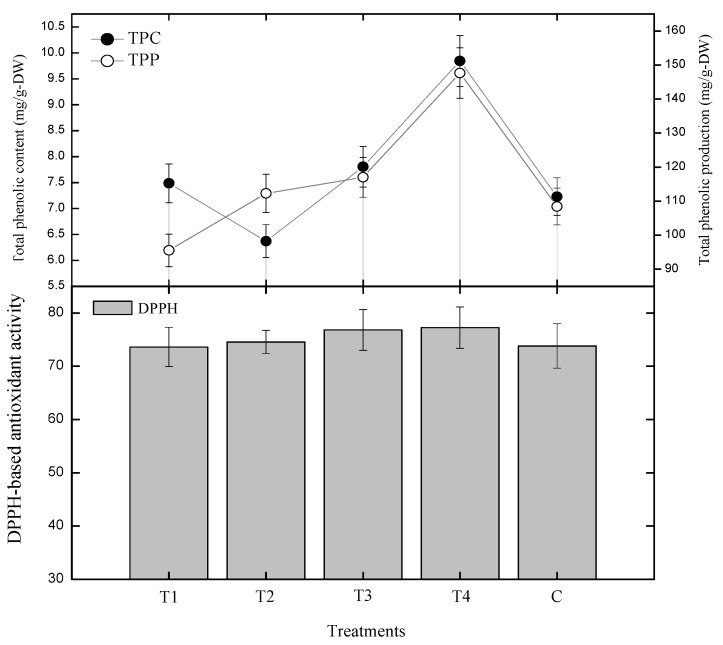
Correlation between DPPH-based antioxidant activity and flavonoids in *S. rebaudiana* adventitious root cultures in vitro. The triplicate data in the mean along with standard errors (SE) are significantly different at *p* < 0.05.

**Figure 6 plants-09-00420-f006:**
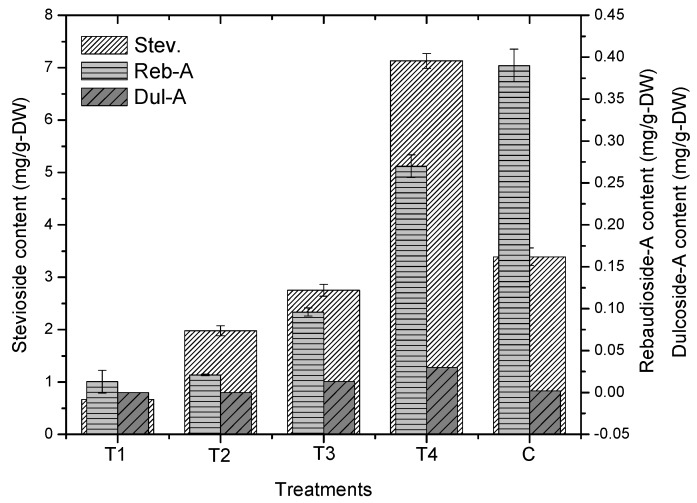
Effect of different GA_3_ concentrations on production of Steviol glycosides. Data were collected from three independent experiments. Triplicate mean values along with standard errors (SE) are significantly different at *p* < 0.05.

**Table 1 plants-09-00420-t001:** Different concentrations of gibberellic acid (GA_3_) applied for biomass accumulation and secondary metabolites production in adventitious root cultures of *Stevia rebaudiana*.

Different Treatments Used in the Present Study		
Treatments	NAA Concentration	GA_3_ Concentration
T1	0.5 mg/L	0.5 mg/L
T2	0.5 mg/L	1.0 mg/L
T3	0.5 mg/L	1.5 mg/L
T4	0.5 mg/L	2.0 mg/L
Control	0.5 mg/L	0.0 mg/L

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
