# Peer review of "Effect of Gibberellic Acid on Production of Biomass, Polyphenolics and Steviol Glycosides in Adventitious Root Cultures of Stevia rebaudiana (Bert.)"

_plants, 2020, doi:10.3390/plants9040420_

Round 1
Reviewer 1 Report
The manuscript is interesting and useful due to the enormous scientific and economic interest associated with the Stevia rebaudiana. Before accepting the manuscript, the authors should make it easier to read: - the manuscript should be read and corrected by a native speaker; -the abbreviations used should be indicated the first time they are used; - bibliographic references in the text and at the bottom of the manuscript should be indicated according to the specifications of the journal; - the figures should be more readable; - correct a series of errors reported in yellow in the attached file

Author Response
Reviewer 1 Comments to Author:
Dear Professor,
Thank you very much for making effort to review our manuscript and your nice suggestions. You have studied our manuscript very carefully and it is kind of you to spend your valuable time on reviewing our manuscript. We followed your nice comments and tried to improve our manuscript according to your suggestions. Hope that the current incorporated suggestions will satisfy your comments.
Comments: Delete (S. rebaudiana).
Response: Dear Professor, we have removed the word (S. rebaudiana) from the first sentence of introduction accordingly.
Comments: Correct the spelling of Asteracae”
Response: Dear Professor, we have corrected the spelling of Asteraceae accordingly”
Comments: Delete. These plants are famous for its sweet taste.
Response: Dear Professor, we have deleted the sentence from introduction accordingly.
Comments: Correct [6]; [5] [7]”
Response: Dear Professor, we have corrected the numbering as [5-7] according to your comments”
Comments: Stevia rebaudiana correct as S. rebaudiana.
Response: Dear Professor, we have converted the full name to abbreviated form as S. rebaudiana.
Comments: Correct [6]; [5] [7]”
Response: Dear Professor, we have corrected the numbering as [5-7] according to your comments”
Comments: Stevia rebaudiana correct as S. rebaudiana.
Response: Dear Professor, we have converted the full name to abbreviated form as S. rebaudiana.
Comments: Replace Gibberellic acid with GA3
Response: Dear Professor, we have Gibberellic acid with GA3 according to your comments”
Comments: Remove GA3.
Response: Dear Professor, we have GA3 from the sentence accordingly.
Comments: Artemisinin “A” should be lowercase
Response: Dear Professor, we have corrected Artemisinin as “artemisinin” according to your comments”
Comments: NAA?.
Response: Dear Professor, we have expanded the abbreviation as naphthalene acetic acid.
Comments: PGRs?.
Response: Dear Professor, we have expanded the abbreviation as plant growth regulators.
Comments: 2.0 mg/l GA3 ensures the best elicitors for root development, why not 1.5 mg/l
Response: Dear Professor, we have incorporated both concentrations (1.5 and 2.0 mg/l) according to your suggestions.
Comments: GAE?.
Response: Dear Professor, we have expanded the abbreviation as Gallic Acid Equivalent (GAE).
Comments: TPC?.
Response: Dear Professor, we have expanded the abbreviation as total phenolics content.
Comments: remove the space between value and unit 73.6 % and 74.5 %
Response: Dear Professor, we have removed the spacing between value and unit as 73.6% and 74.5% according to your suggestions.
Comments: figure 4-5.
Response: Dear Professor, we have corrected it as “figures 4-5).
Comments: 0.5 mg/?.
Response: Dear Professor, we have 0.5 mg/ as “0.5 mg/l”.
Comments: remove the “a” and “b” from figure 2
Response: Dear Professor, we grouped it as a single figure with two different content. The other reviewers also suggests that to explain “a” and “b” in the figure legend. Therefore we explained “a” and “b” in the figure legend as “Biomass and PGRs dependent linear and positive correlation with phenolics biosynthesis (a) and phenolics production (b) in S. rebaudiana adventitious root cultures in vitro”.
Comments: Echinacea Pupurea?.
Response: Dear Professor, we have corrected the botanical name Echinacea pupurea.
Comments: 0.0001 mg/1?.
Response: Dear Professor, we have corrected it “0.0001 mg/1” as “0.0001 mg/l”.
Comments: maize. [37] conducted?
Response: Dear Professor, we have corrected the reference as “maize. Ali et al. [37] conducted”.
Comments: respectively. [35] used?.
Response: Dear Professor, we have corrected the reference as “respectively. Smith et al. [35] used”
Comments: GA3 at 0.01 mg/1?.
Response: Dear Professor, we have corrected it “GA3 at 0.01 mg/1” as “GA3 at 0.01 mg/l”.
Comments: Echinacea purpurea?
Response: Dear Professor, we have replaced the Echinacea by E. purpurea”.
Comments: S. miltiorrhiza [37]. conducted?.
Response: Dear Professor, we have corrected the reference as “S. miltiorrhiza. Ali et al. [37] conducted”
Comments: 2.0mgl stimulate?.
Response: Dear Professor, we have corrected it as “2.0 mg/l” according to your suggestions.
Comments: GA3. [41]. used GA3?
Response: Dear Professor, we have corrected the reference ” GA3. Teszlák et al. [41] used GA3”.
Comments: grape. [36]. studied?.
Response: Dear Professor, we have corrected the reference as “grape. Banyai et al. [36] studied”
Comments: plants. [42] studied?
Response: Dear Professor, we have corrected the reference ” plants. Alonso-Ramírez et al. [42] studied”.
Comments: Arabidopsis. [33] studied?
Response: Dear Professor, we have corrected the reference ” Arabidopsis. Abbasi et al. [33] studied”.
Comments: plant of E. purpura?
Response: Dear Professor, we have corrected the spelling as E. purpurea according to your nice comment”.
Comments: In contrast, [17] observed 10.20 mg/g-DW of stevioside content in Stevia in vitro shoots?.
Response: Dear Professor, we have corrected the sentence as “In contrast, Khalil et al. [17] observed 10.20 mg/g-DW of stevioside content in Stevia in vitro shoots”
Comments: in vitro shoots of Stevia rebaudiana?.
Response: Dear Professor, we have replaced the full name with abbreviated form “S. rebaudiana”
Comments: Furthermore, [46]?
Response: Dear Professor, we have corrected the reference as “Ladygin et al. [46] detected”
Comments: Similarly, [5] and [6]?.
Response: Dear Professor, we have corrected the reference as “Similarly, Mathur and Shekhawat [5] and Reis et al. [6] observed”
Comments: Stevia rebaudiana. Moreover, [47] detected 4.6 mg/g-DW?.
Response: Dear Professor, we have corrected it as ” S. rebaudiana. Moreover, Bondarev et al. [47] detected”
Comments: rebaudioside are sweeter?
Response: Dear Professor, we have corrected the sentence as “rebaudioside is sweeter”
Comments: GA treatments
Response: Dear Professor, we have corrected The GA as GA3 accordingly”
Comments: solidified with 7–8 g l−1 agar?
Response: Dear Professor, we have corrected the sentence as “solidified with 7–8 g/l agar”
Comments: (Sigma; 1.0–10 mg/ml; R2=0.9878)?
Response: Dear Professor, we have corrected it is as (Sigma; 1.0–10 mg/ml; R2=0.9878) accordingly”
Comments: flavonoid content the methodology of [18]. (2014) was?.
Response: Dear Professor, we have corrected the reference as ” flavonoid content the methodology of Fazal et al. [18] was followed”
Comments: (0.25 mg/ 20 ml×4)?
Response: Dear Professor, we have removed the space as ” (0.25 mg/20 ml×4).
Comments: DRSA (%) = 100 x (1- AP /AD)?
Response: Dear Professor, we have removed the space as ” DRSA (%) = 100 x (1- AP/AD).
Comments: protocol of [44]Aman et al. (2013).?
Response: Dear Professor, we have corrected the reference as “protocol of Aman et al. [44].”
Comments: Replace full name with abbreviation?
Response: Dear Professor, we have replaced the Stevia rebaudiana as “ S. rebaudiana” accordingly”
Comments: Table 1A: Different concentrations of GA3 applied for biomass accumulation and secondary metabolites production in adventitious root cultures of Stevia rebaudiana?.
Response: Dear Professor, we have corrected the specie name as ” Table 1A: Different concentrations of GA3 applied for biomass accumulation and secondary metabolites production in adventitious root cultures of Stevia rebaudiana.”
Comments: Correct the journal name in reference 4?
Response: Dear Professor, we have corrected the journal name in ref 4 as “Food Chem Toxicol. 2008, 46 (7), S1-S10”
Comments: Correct the Journal name in ref 5?
Response: Dear Professor, we have corrected the journal name in ref 5 as “Acta Physiol. Plant 2013,”
Comments: Correct the Journal name in ref 6?
Response: Dear Professor, we have corrected the journal name in ref 6 as “Plant Cell Tiss. Org. Cult. 2011,”
Comments: Correct the Journal name in ref 7?
Response: Dear Professor, we have corrected the journal name in ref 7 as “Sugar Tech. 2011”
Comments: Correct the specie name ref 11?
Response: Dear Professor, we have corrected the specie name as “Stevia rebaudiana”
Comments: Correct the specie name ref 13?
Response: Dear Professor, we have corrected the reference 13 as “Curi, R.; Alvarez, M.; Bazotte, R.; Botion, L.; Godoy, J.; Bracht, A. Effect of Stev/A Reba Ud/ANA on glucose tolerance in normal adult humans.. Braz. J. Med. Biol. Res. 1986, 19, 771-774”
Comments: Correct the bacteria name in reference 15?
Response: Dear Professor, we have corrected the specie name as “Escherichia coli”
Comments: Correct the Journal name in ref 17?
Response: Dear Professor, we have corrected the journal name in ref 17 as “Curr. Pharm. Biotechnol. 2016”
Comments: Correct the Journal name in ref 28?
Response: Dear Professor, we have corrected the journal name in ref 28 as “Enzyme Microb. Tech. 1995”
Comments: Correct the Journal name in ref 29?
Response: Dear Professor, we have corrected the journal name in ref 29 as “Curr. Opin Plant Biol. 2003”
Comments: Correct the Journal name in ref 30?
Response: Dear Professor, we have corrected the journal name in ref 30 as “Planta Medica 2005”
Comments: Correct the reference 34?
Response: Dear Professor, we have corrected the ref 34 as “inhibitor on metabolism of tanshinones. Chinese J. Exp. Trad. Med. Formulae 2008”
Comments: Correct the Journal name in ref 46?
Response: Dear Professor, we have corrected the journal name in ref 46 as “Biol. Plant. 2008”
Comments: Correct the ref 48?
Response: Dear Professor, we have corrected the journal name in ref 48 as “Acta. Physiol. Plant 2016, 38, 200, DOI: https://doi.org/10.1007/s11738-016-2218-3”

Reviewer 2 Report
Manuscript Plants-711863 needs major revision before its eventual acceptance:
After line 18, write S. rebaudiana, not Stevia revaudiana.
Subsection 2.3 is too short if compared with subsection 2.2. Authors have forgotten to explain results obtained for rebaudioside-A and dulcoside-A.
The subsection on “Formatting of Mathematical Components” (lines 177-180) should be included in the Materials and Methods section.
Line 186: erase “conducted by”.
Lines 184 and 201: write E. purpurea, not Echinacea purpurea.
Line 240: write “erlenmeyer flasks”, not flasks (Erlenmeyer).
Lines 256-258: it is necessary to indicate the temperature and time used for drying roots. The term “shade” is ambiguous.
Line 267: write “the methodology vy Fazal et al. [18]. Make similar changes in lines 279 and 296.
Line 276: write As, not as.
Subsection 4.4.3 should be rewritten. It is very difficult to understand how the assay was made.
Subsection 4.4.4: please, write rebaudioside-A and dulcoside-A, and describe with detail the instrument Used for HPLC analysis.
It is absolutely necessary that authors clearly indicate the sources of the different reagents used in the experiments. This is of crucial importance in the case of naphthalene acetic acid and gibberellic acid.
Please, use the format of figure 3 for figure 2. In the captions of both figures, the meaning of (a) and (b) should be mentioned.
Please, use GA3, not GA, in the text and figure captions.
Finally, authors should indicate the contributions of three co-authors: Mohammad Ali, Christophe Hano and Bilal Haider Abbasi
Author Response
Reviewer 2 Comments to Author:
Dear Professor,
Thank you very much for your valuable comments. Your comments really impressed me. We are happy for the chance to eliminate these weaknesses in our manuscript. Dear Sir, we specially focused on your comments and it will really modify the revised manuscript. However, we have tried our best to incorporate all of changes/modifications suggested by you
Comments and Suggestions for Authors
Manuscript Plants-711863 needs major revision before its eventual acceptance:
Comments: After line 18, write S. rebaudiana, not Stevia revaudiana.
Response: Dear Professor, we have mentioned the abbreviated name in place of full name in the whole text including, introduction, methodology, results and discussion and table and figure legends according to your suggestions.
Comments: Subsection 2.3 is too short if compared with subsection 2.2. Authors have forgotten to explain results obtained for rebaudioside-A and dulcoside-A.
Response: Dear Professor, we have modified the section 2.3 accordingly. When properly explained the biosynthesis of Rebaudioside-A and Dulcoside-A as” Furthermore, it has been observed that the control treatment displayed maximum rebaudioside-A content which is > 6 mg/g-DW. However, the higher concentration of GA3 (2.0 mg/l) exhibited > 5mg/g-DW biosynthesis of rebaudioside-A content. The other treatments (0.5, 1.0 and 1.5 mg/l GA3) showed lower production of rebaudioside-A content than control and 2.0 mg/l GA3. In this study, the biosynthesis of dulcoside-A content was comparatively lower than stevioside and rebaudioside-A content in adventitious root cultures of Stevia exposed to multiple concentration of GA3. Here, 2.0 mg/l GA3 produced > 1 mg/g-DW of dulcoside-A content which is greater than control and other GA3 treatments. These results suggest that 2.0 mg/l GA3 was found effective for the overall biosynthesis of Steviol glycosides.
Comments: The subsection on “Formatting of Mathematical Components” (lines 177-180) should be included in the Materials and Methods section.
Response: Dear Professor, we have incorporated the “4.6 Formatting of Mathematical Components” in the methodology section according to your suggestion.
Comments: Line 186: erase “conducted by”.
Response: Dear Professor, we have removed the “conducted by” from the discussion section accordingly.
Comments: Lines 184 and 201: write E. purpurea, not Echinacea purpurea.
Response: Dear Professor, we have incorporated the required suggestion and use E. purpurae in place of Echinacea purpurea accordingly
Comments: Line 240: write “erlenmeyer flasks”, not flasks (Erlenmeyer).
Response: Dear Professor, we have corrected the sentence as “Erlenmeyer flasks (100 ml)” according to your valuable comment
Comments: Lines 256-258: it is necessary to indicate the temperature and time used for drying roots. The term “shade” is ambiguous.
Response: Dear Professor, thank you for this useful comment, we have corrected the sentence as “these roots were dried in oven at 50 °C for 24 h”
Comments: Line 267: write “the methodology vy Fazal et al. [18]. Make similar changes in lines 279 and 296.
Response: Dear Professor, we have corrected the references as “The determination of TPC of each sample was carried out according to the methodology of Fazal et al. [18].” And, “methodology of Fazal et al. [18] was followed for flavonoids” and for quantification of Steviol glycosides we used the protocol of Aman et al. [44].”
Comments: Line 276: write As, not as.
Response: Dear Professor, we have corrected the “as” as “As” as In the above equation the “As” stands according to your suggestions.
Comments: Subsection 4.4.3 should be rewritten. It is very difficult to understand how the assay was made.
Response: Dear Professor, we have tried our level best to simplify the methodology of antioxidant activity as “The extract was prepared by adding 5 mg of treated cultures powdered in 20 ml HPLC grade ethanol. Exactly, 1 ml of treated culture extract was mixed with 2.0 ml of DPPH free radical solution. The DPPH free radicals powdered (0.25 mg) was dissolved in 20 ml HPLC grade ethanol (Sigma). Followed by incubation for 30 min in the dark. The absorbance of resulted mixture was checked at room temperature at 517 nm to obtain OD below than 1.0. If the OD is greater than 1 then it need 4 time dilution? The mixture of treated culture (1 ml) and DPPH solution (2 ml) was added to cuvette of spectrophotometer (UV-visible double beam Spectrophotometer; London) and radical scavenging activity was calculated as percentage of DPPH discoloration using the following equation at 517 nm”
Comments: Subsection 4.4.4: please, write rebaudioside-A and dulcoside-A, and describe with detail the instrument Used for HPLC analysis.
Response: Dear Professor, we have tried our level best to modify this section according to your valuable comments as “S. rebaudiana adventitious root cultures were investigated for stevioside, rebaudioside-A and dulcoside-A contents by using the protocol of Aman et al. [44]. For Steviol glycoside quantification we used Perkin-Elmer HPLC system (USA). The setting including C18 column, vacuum degasser, quaternary pump, variable wavelength detector, and injection loop with 10-μl capacity. For mobile phase HPLC grade water (Sigma; A; 25%) and acetonitrile (Sigma; B; 75%) was used. During quantification, volume of 10 µl was used to inject at 1.0 ml/min flow rate. Steviol glycoside standard containing stevioside, rebaudioside-A and dulcoside-A, purchased from Sigma (USA) laboratories was run at first for standardization of retention time of each content. Stevioside, rebaudioside-A and dulcoside-A contents were identified in each sample of adventitious root cultures by comparing retention times of samples with standard. The diterpene glycosides (dulcoside-A, rebaudioside-A and stevioside) were indicated in the results as mg per g on the basis of dry weight.”
Comments: It is absolutely necessary that authors clearly indicate the sources of the different reagents used in the experiments. This is of crucial importance in the case of naphthalene acetic acid and gibberellic acid.
Response: Dear Professor, we have incorporated the required suggestions as
“Murashige and Skoog medium (MS; 1962; Phyto Technologies Laboratories, USA)”
“naphthalene acetic acid (0.5 mg/l NAA; Phyto Technologies Laboratories, USA)”
“Gibberellic acid (GA3; Phyto Technologies Laboratories, USA)”
We also mentioned the source of other chemical like ethanol etc.
Comments: Please, use the format of figure 3 for figure 2. In the captions of both figures, the meaning of (a) and (b) should be mentioned.
Response: Dear Professor, we have corrected the figure 2 and 3 captions as
“Figure 2: Biomass and PGRs dependent linear and positive correlation with phenolics biosynthesis (a) and phenolics production (b) in S. rebaudiana adventitious root cultures in vitro. The triplicated data in mean along with standard errors (SE) are significantly different at P < 0.05.”
“Figure 3: Biomass and PGRs dependent linear and positive correlation with flavonoids biosynthesis (a) and flavonoids production (b) in S. rebaudiana adventitious root cultures in vitro. The triplicated data in mean along with standard errors (SE) are significantly different at P < 0.05.”
Comments: Please, use GA3, not GA, in the text and figure captions.
Response: Dear Professor, we have replaced the entire GA with “GA3” in the whole text and figure captions accordingly
Comments: Finally, authors should indicate the contributions of three co-authors: Mohammad Ali, Christophe Hano and Bilal Haider Abbasi
Response: Dear Professor, we have added the contribution as “Bilal Haider Abbasi wrote the whole manuscript and Christophe Hano review the final manuscript for journal submission.” According to your suggestions

Round 2
Reviewer 2 Report
The revised version of manuscript Plants-711863 needs minor revision before its eventual acceptance:
Certainly, subsection 4.4.3 has been improved. Unfortunately, authors should revised it, because still it is difficult to understand.
Subsection 4.4.4: authors should describe with detail the instrument used for HPLC analysis (type of column, pump and detector models, software used for controlling the system and for obtaining data...). For instance, there are dozens, even hundreds, of different C18 columns available in the market. The information included is insufficient to reproduce the experiments.
Author Response
Reviewer 2 Comments to Author:
Dear Professor,
Thank you very much for making effort to review our manuscript and your nice suggestions. You have studied our manuscript very carefully and it is kind of you to spend your valuable time on reviewing our manuscript. We followed your nice comments and tried to improve our manuscript according to your suggestions. Hope that the current incorporated suggestions will satisfy your comments.
Comments and Suggestions for Authors
The revised version of manuscript Plants-711863 needs minor revision before its eventual acceptance:
Comments: Certainly, subsection 4.4.3 has been improved. Unfortunately, authors should revised it, because still it is difficult to understand.
Response: Dear Professor, According to your nice suggestions we have revised this section as “The methodologies of Ahmad et al. [24] and Zamir et al. [48] were followed for determination of DPPH radical scavenging activity (DRSA). The DRSA activity was determined for each treatment of GA3. The DRSA need 2 stock solutions preparation. The first stock solution is the extract preparation from various adventitious roots culture exposed to various GA3 concentrations (0.5, 1.0, 1.5, 2.0 mg/l). The dry adventitious roots powdered (5 mg) exposed to 0.5 mg/l GA3 was separately dissolved in 20 ml HPLC grade ethanol (Sigma Aldrich, Germany). The solution was centrifuged at 10,000 rpm for 10 min (PV-1; Grant Instruments, UK). The supernatant was collected for further experiment. The same procedure was followed for other treatments of GA3. The second stock solution is the DPPH free radicals, in which DPPH free radicals powdered (Sigma Aldrich; 0.25 mg) was dissolved in 20 ml HPLC grade ethanol (Sigma Aldrich, Germany). It is necessary to check the OD of the DPPH solution at 517 nm using spectrophotometer (UV-visible double beam Spectrophotometer; London). If the OD of the DPPH solution is greater than 1.0 then it need further dilution. Add 20 ml ethanol and again check its OD. If the OD is below than 1.0 then it is ready for antioxidant activity. The final step is the investigation of antioxidant activity. Here, 1 ml solution was taken from stock 1 (GA3 treated culture) and combined it with 2 ml of stock 2 (DPPH solution) in spectrophotometer cuvette. To avoid oxidation and to complete the reaction, the combined solution was incubated for 30 min in dark. After incubation, the absorbance was checked at 517 nm and radical scavenging activity was calculated as percentage of DPPH discoloration using the following equation at 517 nm”
Comments: Subsection 4.4.4: authors should describe with detail the instrument used for HPLC analysis (type of column, pump and detector models, software used for controlling the system and for obtaining data...). For instance, there are dozens, even hundreds, of different C18 columns available in the market. The information included is insufficient to reproduce the experiments.
Response: Dear Professor, we have modified this section as “S. rebaudiana adventitious root cultures were investigated for stevioside, rebaudioside-A and dulcoside-A contents by using the protocol of Aman et al. [44]. To prepare extract for HPLC analysis, 200 mg dried powdered from each treatment was dissolved independently in 10 ml HPLC grade ethanol. The extraction was repeated for three times. After 72 h, the ethanol was evaporated under control pressure at 40 °C. After ethanol evaporation, a crude extract was obtained. Exactly 1 mg of extract from each treatment was independently dissolved in 2 ml of ethanol. The solution was filtered through 0.45 Millipore micro-filter. The filtrate was then used for quantification of Steviol glycoside through HPLC. For Steviol glycoside quantification we used Perkin-Elmer HPLC system (USA). The HPLC conditions includes, HPLC equipped with a binary HPLC pump (1525), the column was C18 (ODS) with a 25×4.6-mm, 5-μm particle size and with 5 mm internal diameter, and 2487duel λ absorbance detector attached with the integrated Empower 2 software. The analysis was performed at ambient temperature with injection loop with 10-μl capacities. For mobile phase HPLC grade water (Sigma; A; 25%) and acetonitrile (Sigma; B; 75%) was used. During quantification, volume of 10 µl was used to inject at 1.0 ml/min flow rate. Steviol glycoside standard containing stevioside, rebaudioside-A and dulcoside-A, purchased from Sigma (USA) laboratories was run at first for standardization of retention time of each Steviol glycoside. Stevioside, rebaudioside-A and dulcoside-A contents were identified in each sample of adventitious root cultures by comparing retention times of samples with standard. The diterpene glycosides (dulcoside-A, rebaudioside-A and stevioside) were indicated in the results as mg per g on the basis of dry weight.”
